# Association of Clonal Hematopoiesis of Indeterminate Potential with Inflammatory Gene Expression in Patients with COPD

**DOI:** 10.3390/cells11132121

**Published:** 2022-07-05

**Authors:** Stefan Kuhnert, Siavash Mansouri, Michael A. Rieger, Rajkumar Savai, Edibe Avci, Gabriela Díaz-Piña, Manju Padmasekar, Mario Looso, Stefan Hadzic, Till Acker, Stephan Klatt, Jochen Wilhelm, Ingrid Fleming, Natascha Sommer, Norbert Weissmann, Claus Vogelmeier, Robert Bals, Andreas Zeiher, Stefanie Dimmeler, Werner Seeger, Soni S. Pullamsetti

**Affiliations:** 1University of Giessen and Marburg Lung Center (UGMLC), German Center for Lung Research (DZL), Cardio-Pulmonary Institute (CPI), Justus Liebig University, 35392 Giessen, Germany; stefan.kuhnert@innere.med.uni-giessen.de (S.K.); rajkumar.savai@mpi-bn.mpg.de (R.S.); stefan.hadzic@med.uni-giessen.de (S.H.); jochen.wilhelm@chemie.bio.uni-giessen.de (J.W.); natascha.sommer@innere.med.uni-giessen.de (N.S.); norbert.weissmann@innere.med.uni-giessen.de (N.W.); werner.seeger@innere.med.uni-giessen.de (W.S.); 2Max Planck Institute for Heart and Lung Research, DZL, CPI, 61231 Bad Nauheim, Germany; siavash.mansouri@mpi-bn.mpg.de (S.M.); edibe.avci@mpi-bn.mpg.de (E.A.); gabrieladiazpi@gmail.com (G.D.-P.); manjupadma-sekar.nandigama@mpi-bn.mpg.de (M.P.); mario.looso@mpi-bn.mpg.de (M.L.); 3Department of Medicine, Hematology/Oncology, University Hospital Frankfurt, CPI, Goethe University, 60596 Frankfurt am Main, Germany; m.rieger@em.uni-frankfurt.de; 4Frankfurt Cancer Institute (FCI), CPI, Goethe University, 60596 Frankfurt am Main, Germany; 5Institute for Lung Health (ILH), Justus Liebig University, 35392 Giessen, Germany; 6Institute for Neuropathology, CPI, Justus Liebig University, 35392 Giessen, Germany; till.acker@patho.med.uni-giessen.de; 7Institute of Vascular Signalling, Department of Molecular Medicine, CPI, Goethe University, 60596 Frankfurt am Main, Germany; klatt@vrc.uni-frankfurt.de (S.K.); fleming@vrc.uni-frankfurt.de (I.F.); 8Department of Medicine, Pulmonary and Critical Care Medicine, Philipps University of Marburg, DZL, 35043 Marburg, Germany; clausfranz.vogelmeier@uk-gm.de; 9Department of Internal Medicine V-Pulmonology, Allergology and Critical Care Medicine, Saarland University, 66421 Homburg, Germany; robert.bals@uks.eu; 10Department of Medicine, Cardiology, CPI, Goethe University Hospital, 60596 Frankfurt am Main, Germany; zeiher@em.uni-frankfurt.de; 11Institute for Cardiovascular Regeneration, CPI, Goethe University, 60596 Frankfurt am Main, Germany; dimmeler@em.uni-frankfurt.de

**Keywords:** clonal hematopoiesis, inflammation, COPD

## Abstract

Chronic obstructive pulmonary disease (COPD) is a disease with an inflammatory phenotype with increasing prevalence in the elderly. Expanded population of mutant blood cells carrying somatic mutations is termed clonal hematopoiesis of indeterminate potential (CHIP). The association between CHIP and COPD and its relevant effects on DNA methylation in aging are mainly unknown. Analyzing the deep-targeted amplicon sequencing from 125 COPD patients, we found enhanced incidence of CHIP mutations (~20%) with a predominance of DNMT3A CHIP-mediated hypomethylation of Phospholipase D Family Member 5 (*PLD5*), which in turn is positively correlated with increased levels of glycerol phosphocholine, pro-inflammatory cytokines, and deteriorating lung function.

## 1. Introduction

Accumulation of somatic mutations throughout human life is a clearly defined phenomenon becoming more pronounced by aging. Hematopoietic stem cells (HSCs) with extensive self-renewal capacity are one of the most vulnerable cells to various somatic mutation with 70 mutations per gene for one HSC by age of 70 [1]. Different studies have shown that expansion of mutated HSCs, known as “Clonal hematopoiesis (CH)” is highly increased by aging [2]. CHIP has been described previously to be a driver of altered inflammatory response in chronic diseases, e.g., cardiovascular disease [3]. Age and smoking are the two strongest risk factors of chronic obstructive pulmonary disease (COPD), which contribute to the aberrant inflammatory phenotype of the disease [4]. In a very recent study, CHIP was shown to be associated with disease development and severity of COPD from four different cohorts. Miller et al. identified the increased risk of COPD related to CHIP independently of age, cigarette smoke exposure, and inherited polygenic risk score [5]. In other aspects, the association between CHIP and COPD and more importantly, its relevant effects on DNA methylation as one of the key players in aging are mainly unknown. In this study, we aimed (I) to investigate the prevalence of CHIP in COPD patients, (II) to investigate CHIP-mediated molecular consequences by DNA methylation, metabolomics, and cytokine analyses, (III) to correlate these with lung function parameters, and (IV) to evaluate the causality in macrophages derived from human peripheral blood mononuclear cells (PBMCs) and human precision cut lung slides (PCLS).

## 2. Materials and Methods

### 2.1. Study Population

Data were obtained from the baseline examination (Visit 1) of the German multicenter COPD cohort COSYCONET (German COPD and Systemic Consequences–Comorbidities Network). COPD was defined according to the GOLD criteria after performing standardized post-bronchodilator spirometry. COSYCONET is a German multicenter prospective observational trial, which recruited 2741 patients aged 40 years and older with diagnosis of COPD between 2010 and 2013 in 31 study centers. The study protocol has been previously described in detail [6] and in the Appendix A.

### 2.2. DNA Methylation

DNA was isolated from the EDTA blood samples obtained from COSYCONET. The Infinium Human-Methylation EPIC BeadChip (850k) (WG-317, Illumina, San Diego, CA, USA) was used to determine the DNA methylation status following the producer’s guidelines. Infinium human methylation EPIC array analysis was performed by using RnBeads2.0, Max Planck Institute for Heart and Lung Research, release 2.0. https://rnbeads.org (accessed on 5 July 2019) [7] and ADMIRE, Max Planck Institute for Heart and Lung Research, release 2.0 https://github.molgen.mpg.de/loosolab/admire (accessed on 5 July 2019) [8]. More information regarding DNA isolation and methylation analysis has been provided in the Appendix A.

### 2.3. Sample Preparation for NGS and High-Throughput Sequencing

Samples of 125 COPD patients were assessed for the presence of CHIP by the Illumina TruSeq Custom Amplicon Low Input assay. The panel includes 594 amplicons in 56 genes commonly mutated in CHIP and myeloid malignancies (Appendix A) [9]. The pooled libraries were sequenced on a NextSeq 500 sequencer (Illumina) using the NextSeq 500/550 Mid Output, version 2 kit (300 cycles) according to the manufacturer’s instructions. The procedure has been described in detail in the Appendix A.

### 2.4. Metabolomics and Cytokine Analysis

Oxylipins and choline pathway metabolites were extracted from plasma by adding 3× the volume of ethyl acetate and 4× the volume of methanol, respectively. Either the upper layer (oxylipins) or supernatant (cholines) was analyzed on an Agilent 1290 Infinity LC system coupled to a Sciex QTrap 5500 mass spectrometer in negative (oxylipins) or positive (cholines) ionization modes. TNF-α and IL-6 were analyzed using ELISA kits. The detailed protocol for plasma collection and metabolome analysis has been explained in the Appendix A.

### 2.5. Cell Culture and Gene Expression Analysis by qPCR

Generation of human macrophages from PBMCs, RNA isolation, complementary DNA synthesis, and quantitative PCR were performed as previously described [10]. The brief information has been provided in the Appendix A.

### 2.6. Human COPD Precision Cut Lung Slides (PCLS)

PCLS were cut 400 µm thick using a vibratome (Microm HM650V, Thermo Fisher Scientific Inc., Waltham, MA, USA) and cultured in growth media. PCLS were then treated with conditional media of a DNMT3A-deficient monocyte cell line (THP1) for 48 h. The tissue lysates were taken for protein expression studies. Detailed protocol has been explained in the Appendix A.

### 2.7. Western Blotting

The tissues were lysed in RIPA lysis buffer containing protease and phosphatase inhibitors. Subsequent step have been described in detail in the Appendix A.

### 2.8. Statistics

All analyses were carried out with the statistics software R v3.4 https://www.R-project.org (accessed on 5 July 2019). Lung function data were analyzed with general linear models using the log-transformed response values, and p values were adjusted by confounding factors including age, sex, smoking status, and pack years.

## 3. Results and Discussion

In this study, we investigated the prevalence of CHIP in a COPD patient cohort (COSYCONET) (Table 1).

Deep targeted amplicon sequencing in the patients showed an enhanced incidence of CHIP mutations compared with age-matched controls (~20%), with the striking predominance of somatic mutations in the gene *DNMT3A* (18 out of 25) (Figure 1A). The same profile was detected in early-stage COPD (Gold 0). In-depth analysis of the mutations and in silico analysis via Swiss Model https://swissmodel.expasy.org/ (accessed on 5 July 2019) identified 6 mutations (from 20) in *DNMT3A* which significantly alter its structure, suggesting reduced or even loss of methyl-transferase activity (Appendix A, In silico analysis of DNMT3A and Appendix A). Interestingly, DNA methylation analysis demonstrated massive hypomethylation in DNMT3A-CHIP COPD versus Non-CHIP COPD patients, in up to 95% of top differentially methylated CpG islands, and 80% of differentially methylated CpG sites and tilings (Figure 1B).

As a possible indicator of CHIP mutations [3], we evaluated differences in proinflammatory cytokine formation and lung function between CHIP-positive and CHIP-negative early COPD patients. In early-stage COPD patients, significant higher levels of circulating inflammatory mediator IL6 and slight increase of TNFα were indeed noted in the CHIP mutation carriers, concomitant with increased functional residual capacity and decreased PaCO_2_ levels, indicating ventilation/perfusion mismatch and forced ventilatory drive (Figure 1C,D). The numbers of CHIP-positive early COPD patients were too low to link these findings to the smoking history of these patients.

To prove causality, DNMT3A was silenced in non-stimulated macrophages derived from human PBMCs, a significant resulting in an upregulation of pro-inflammatory cytokines, especially *IL6* and slightly *TNFα* (Figure 1E). Human PCLS cultured with conditional media of DNMT3A deficient-THP1 confirmed a link between DNMT3A dysfunction and enhancement of pro-inflammatory (IL-1ß and TNFα) responses (Figure 1F–H). Nevertheless, the DNA methylation status of inflammatory cytokines was not altered (Appendix A), raising the intriguing possibility of additional molecular players driving the pro-inflammatory gene expression. The top hypomethylated tiling of genes includes Transmembrane Protein 232 (*TMEM232*), Glycophorin A (*GYPA*), and Follistatin Like 4 (*FSTL4*), which were previously shown to be associated with COPD and inflammation [11] (Figure 2A).

Notably, one of the top hypomethylated gene families that could be associated with both inflammation and COPD is Phospholipase D Family Member 5 (*PLD5*). We observed that in DNMT3A-CHIP COPD patients, *PLD5* is hypomethylated (Figure 2B) and positively correlated with PaCO_2_ levels of DNMT3A-CHIP COPD, a phenomenon that was not observed in non-CHIP COPD patients (Figure 2C,D). Importantly, *PLD5* methylation levels negatively correlated with inflammatory cytokines in CHIP-positive COPD patients, but not in CHIP-negative patients (Figure 2E,F), confirming that *PLD5*-mediated metabolic pathway modulates systemic oxidative stress and inflammation [12].

Phospholipase D isoforms are lipid-signaling enzymes that preferentially hydrolyze phosphatidylcholine (PC), thus generating phosphatidic acid and choline. Phosphatidic acid is further metabolized to either diacylglycerol (DAG) by phosphatidate phosphohydrolase (lipid phosphate phosphohydrolase) or lysophosphatidic acid (LPA) by phospholipase A_1_/A_2_ (PLA). PLAs are the rate-limiting enzyme for the downstream synthesis of prostaglandins and leukotrienes that are the main mediators of inflammation [12]. Importantly, previous studies confirmed the involvement of PLAs in the COPD-associated inflammation [13] although there are some gaps related to the regulation of specific enzymes in this pathway. To explore whether DNMT3A-CHIP mutations may explain these underlying molecular mechanisms, we performed a metabolome analysis of arachidonic acid-related metabolites as well as choline-containing metabolites in human plasma obtained from DNMT3A-CHIP COPD and Non-CHIP COPD patients. Notably, higher levels of glycerol phosphocholine were observed in DNMT3A-CHIP COPD versus Non-CHIP COPD patients (Figure 2G,H), which are negatively correlated with *PLD5* methylation levels.

We conclude that somatic mutations in hematopoietic cells, specifically in the most commonly mutated CHIP driver gene *DNMT3A*, are increased in COPD and may be associated with metabolic and inflammatory sequelae with the functional outcome (Figure 2I). Underlying mechanisms such as smoking or the lung disease itself may enhance the CHIP frequency in the bone marrow stem cell niche, which then—as a vicious feedback loop—may amplify the pulmonary pathogenetic sequelae. Long-term observations on the impact of CHIP on the course of COPD are mandatory.

## Figures and Tables

**Figure 1 cells-11-02121-f001:**
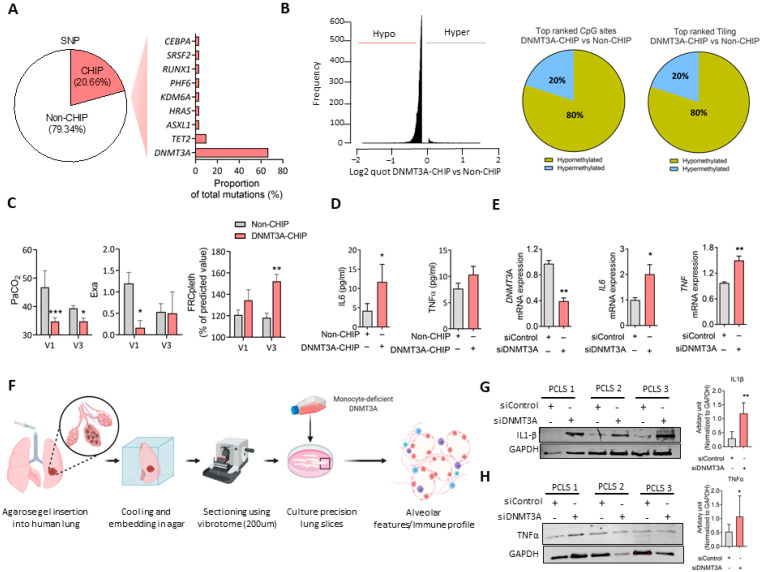
Clonal hematopoiesis of indeterminate potential (CHIP) and mutations in DNMT3A gene is prevalent in COPD patients. (**A**) The % of CHIP mutation carriers and the proportion of CHIP-associated mutations (in 56 analyzed genes) in patients with COPD is given. (**B**) Distribution of hypomethylated (Hypo) and hypermethylated (Hyper) CpG sites in DNMT3A-CHIP and Non-CHIP patients as well as the depiction of the methylation status of top-ranked CpGs and tiling sites in DNMT3A-CHIP COPD and non-CHIP COPD patients. (**C**) Lung function parameters FRCpleth, Exa and PaCO_2_ in 26 early-stage patients with CHIP mutation (CHIP; *n* = 6) or without CHIP mutation (Non-CHIP; *n* = 20), observed over 1.5 years, are given. V1—first visit upon entry into the COSYCONET cohort; V3—third visit after 18 ± 1 months. (**D**) Plasma IL6 and TNFα levels in CHIP (*n* = 6 for IL6 and TNFα) and Non-CHIP patients (*n* = 19 for IL6 and TNFα) with early-stage COPD. *p* values are adjusted by confounding factors including age, sex, smoking status, and pack years. (**E**) mRNA expression of *DNMT3A*, *TNF**α*, and *IL6* in PBMC-derived monocytes, that were subsequently differentiated into macrophages and are transfected with siControl (non-silencing control siRNA) and siDNMT3A for 48 h (*n* = 3 independently performed experiments) to mimic the hypomethylation status observed by *DNMT3A* SNPs in COPD, 2-tailed Student’s *t* test. (**F**) Schematic presentation of ex vivo cultured human PCLS with conditional media of DNMT3A deficient monocyte cell line (THP1), that was transfected with siControl (non-silencing control siRNA) and siDNMT3A for 48 h. (**G**,**H**) Representative western blotting images of IL1β and TNFα followed by densitometric quantification of relative expression IL1β and TNFα with GAPDH as the loading control in PCLS (*n* = 5), 2-tailed Student’s *t*-test. *p* values are adjusted by confounding factors including age, sex, smoking status, and pack years. PaCO_2_—Partial pressure of carbon dioxide in mmHg. FRCpleth—functional residual capacity measured by body plethysmography. Exa—exacerbations. *: *p* < 0.05, **: *p* < 0.01, ***: *p* < 0.001.

**Figure 2 cells-11-02121-f002:**
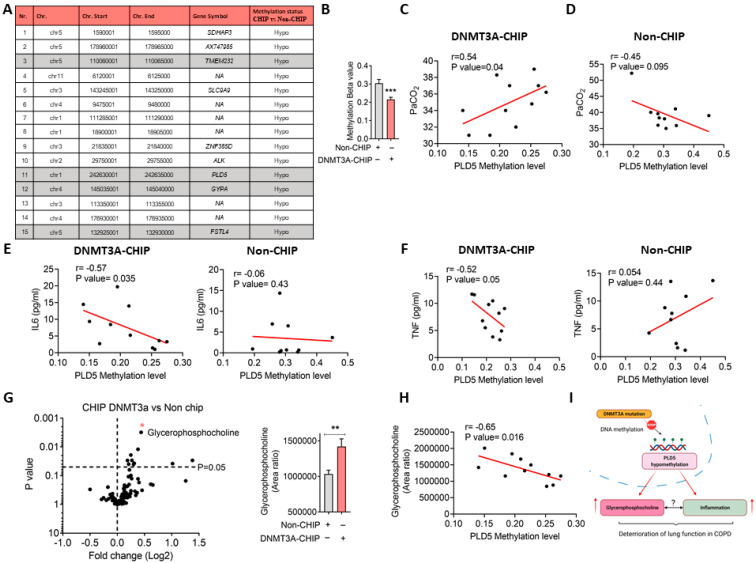
PLD5 gene methylation association with pro-inflammatory signaling and impaired lung function in early COPD. (**A**) Characteristics of top 15 differentially methylated tiling sites between DNMT3A-CHIP and Non-CHIP patients. (**B**) Methylation level of tiling-associated PDL5 between DNMT3A-CHIP and Non-CHIP patients, ***: *p* < 0.001, 2-tailed Student’s *t* test. (**C**,**D**) Pearson correlation analysis between *PLD5* methylation and PaCO_2_ level in DNMT3A-CHIP (N = 11) and Non-CHIP (N = 10) patients. (**E**) Pearson correlation analysis between *PLD5* methylation and plasma IL6 concentration in DNMT3A-CHIP (N = 11) and Non-CHIP (N = 10) patients. (**F**) Pearson correlation analysis between *PLD5* methylation and plasma TNF concentration in DNMT3A-CHIP (N = 11) and Non-CHIP (N = 10) patients. (**G**) Volcano plot of metabolome analysis of serum in DNMT3A-CHIP (N = 18) and Non-CHIP (N = 22) patients. **: *p* < 0.01, 2-tailed Student’s *t* test (**H**) Pearson correlation analysis between *PLD5* methylation and serum glycerolphosphocholine concentration in DNMT3A-CHIP (N = 11). (**I**) Schematic diagram displaying the link between DNMT3A dysfunction and enhancement of metabolic and pro-inflammatory responses in COPD.

**Table 1 cells-11-02121-t001:** Detailed characteristics of CHIP in COPD patient cohort (COSYCONET).

Subset		GOLD 0 Patients			GOLD I–III Patients			GOLD III/IV Patients		
CHIP		All	Negative	Positive	All	Negative	Positive	All	Negative	Positive
Number		26	20	6	90	71	19	36	26	10
Sex	Male	42%	30%	83%	68%	69%	63%	61%	58%	70%
	Female	58%	70%	17%	32%	31%	37%	39%	42%	30%
Age		60 (5)	59 (4.7)	62 (5.6)	62 (5.5)	62 (5.8)	62 (4.4)	62 (5.7)	62 (6.1)	62 (4.5)
BMI		27 (4.8)	26 (5.4)	28 (2.1)	27 (5.8)	27 (6.2)	26 (4.1)	26 (4.7)	27 (5.2)	25 (3.2)
Smoking	Packyears	37 (19)	34 (17)	46 (26)	42 (31)	41 (31)	46 (29)	48 (29)	48 (28)	50 (33)

## Data Availability

Not applicable.

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
