# Peer review of "Association of Clonal Hematopoiesis of Indeterminate Potential with Inflammatory Gene Expression in Patients with COPD"

_cells, 2022, doi:10.3390/cells11132121_

Round 1
Reviewer 1 Report
This is a very well written manuscript with interesting novel findings on the prevalence of Clonal Hematopoiesis of Indeterminate Potential (CHIP) and DNMT3A gene mutation in COPD patients. The authors have used multiple approaches to determine the possible role of inflammatory cytokines as prognostic/ indicating markers of CHIP mutations. Specifically, the correlation of Inflammatory signatures, hypomethylated genes (DNMT3A) and lung functions with CHIP-positive COPD patients is very novel. The sample size and the statistical analysis is very robust.
Author Response
R1: We would like to thank the reviewer for appreciating our efforts in determining the CHIP mutations in COPD patients, the contribution of DNMT3A mutation-mediated DNA methylation changes and their correlation to Inflammatory signatures, hypomethylated genes (DNMT3A) and lung functions with CHIP-positive COPD patients.
Reviewer 2 Report
In the current brief report titled ’Association of Clonal Hematopoiesis of Indeterminate Potential with Inflammatory Gene Expression in Patients with COPD’, Kuhnert S. and coworkers explores the presence of clonal hematopoiesis of indeterminate potential and its association with the gene expression of inflammatory markers in patients with chronic obstructive pulmonary disease (COPD). In 125 patients from the COSYCONET cohort among the CHIP – genes, they find variants in the DNMT3A gene to be the most prevalent and linked to hypomethylation in COPD patients compared to non-COPDs. Analysis of IL6 and TNF levels with respect to DNMT3A variants leads to investigating the gene expression levels of the cytokines when modulating DNMT3A activity by siRNA and subsequent analysis of protein expression levels of IL1 with similar inhibition. Absence of change in methylation status of the cytokines, lead the investigators onwards to examine gene methylation patterns in other genes and discover the PLD5 gene to be hypomethylated. The methylation level appears associated with the partial pressure of carbon dioxide, serum IL6 -and TNF levels for DNMT3A CHIP patients, but not for non-CHIPS. To investigate the association of choline containing metabolites downstream from PDL5 in the CHIP vs non-CHIP groups, serum plasma levels of GPC are measured by mass spectrometry revealing increase with DNMT3A-CHIP and inverse correlation with PLD5 gene methylation.
Recently it has become evident that clonal hematopoiesis is associated with COPD, where DNMT3A appears to be particularly prevalent and is associated with an aberrant inflammatory state, making the study by Kuhnert and co-workers to further shed light upon the field, timely and highly relevant. The study includes analysis of patient clinical data and data from materials both on DNA, RNA and protein levels, in addition to functional cellular studies, which is very satisfactory.
However, there are several issues that must be addressed before the manuscript is suitable for publication.
1. In the introduction – or at least in the discussion, the authors should quote earlier works of COPD and CHIP. As the format for the brief report must be executed in strict confines, one such reference could be the recent, comprehensive analysis by Miller PG, Blood 2022.
2. In the materials and methods, more information regarding the techniques should be included.
a. Section 2.1: More information about the COSYCONET is needed, as a minimum information about patient consent, Helsinki declaration and ethics approval for the project. Furthermore it should be stated if this is a cross-sectional cohort – and if so, how the authors handle patients with myeloproliferative disorders (AML, MDS and MPNs – that are known carriers of the CHIP mutations).
b. Section 2.3: The NGS description could benefit from a short description of NGS pipe-line such as read-depths and filtration, in addition to software and metrics for variant calling.
c. Section 2.4: The authors should include information on plasma was isolated and stored.
d. Section 2.5: The qPCR method should be briefly recapitulated (reference genes etc).
e. Section 2.6: Information regarding Western blotting is missing.
f. Section 2.7: The specific statistical tests for used for the datasets should be specified – either in this section or in the figure text, where relevant.
3. In the manuscript the authors definition of CHIP should be included. Is there a cut-off for the mutant allele fraction derived from NGS? Are any diagnoses excluded?
4. If available, the allele fraction of DNMT3A should be coupled to the GOLD subsets and the remaining analysis of the manuscript.
5. Line 90 – the in silico analysis of the variants should be explained. Is this FATHMM predictions or information from COSMIC, CLINVAR or LOVD databases?
6. Line 120-121 states that COPD have higher levels of inflammatory mediators (IL6, TNF), however, this was only significant for IL6 and not TNF (1D).
7. Line 130 states that the methylation status of the inflammatory cytokines was unaltered – where is this shown?
8. Table 1. The GOLD scale is usually partitioned in I- IV with GOLD IV being <30% FEV1. It is unclear what the GOLD 0 population is, why the GOLD III is separated into two groups and if the GOLD IV is among these – the authors should elaborate. Furthermore, it is unclear whether the number in brackets after Age, BMI and Pack years represent SD.
9. Figure 1.
a. In figure 1B a shift is seen for the methylation for the DNMT3A-CHIP vs non-CHIP. How great is this shift (fold) and is it statistically significant?
b. In figure 1C, D and E - the number (n) should be shown on the graph or be more clearly explained in the figure text. Also, considering that the siRNA experiments are qPCR data the error bars are very well behaved. Do they represent three independently performed experiments with silencing, which they should, or three replicates?
c. Non-CHIP, DNMT3A +/- chip below the x-axis for 1D and E is confusing and suggests a functional interference/addition – like in fig 1 E. It is sufficient to just call them either DNMT3a CHIP or Non-CHIP. Furthermore, as there are no p values on the graphs, the 0.09 above TNF alpha is irrelevant, but merely shows the test did not reach significance.
d. In fig 1E although the neat variance between replicates in IL6 and TNF expression is differs statistically, the authors should comment on the very low relative change <2 fold (especially for TNF) and the validity in context of the technical uncertainties for qPCR testing.
e. Figure 1 F is a very nice overview, but is too small to clearly see. It is furthermore very nice that the authors supplement RNA expression levels with protein expression levels, but the change from IL6 and TNF to IL1-B needs to be justified. All are valid markers, but but should mirror each other in the two assays. The Western blot is not described in the figure text and how was the quantification performed? In the figure text, the p values could be in a single entry at the end instead of repetitions though the text.
10. Figure 2.
a. Figure 2A. The table is too small to clearly see.
b. How was PDL5 chosen from the other hypomethylated genes? Because of methylation beta value? Have the authors ensured this is not a chance finding?
c. In 2C and 2D the message is potentially too skewed by the single outlier value at very high PaCO2 in the non-CHIP population – and without it, the interpretation might change.
d. In fig 2E for the non-CHIP population are the very low values for IL6 within the linear range for quantification? Was the general cytokine levels in these samples sufficiently high?
e. In 1 H, how did the methylation levels correlate in the non-CHIP population
f. When comparing CHIP vs non-CHIP populations, the graphs should have similar y-axis as seen for 1F.
11. In the reference section, there seems to have occurred an error in the numbering (they are repeated).
Author Response
Responses to reviewer 2 are provided in the attached docx file.

Reviewer 3 Report
In this manuscript, Kuhnert et al. reported a study of chronic obstructive pulmonary disease, focused on the clonal hematopoiesis observed in COPD patients and the relationship to the inflammatory profile.
Overall, this study is designed fine. The results indicated potential mechanisms between the inflammatory profile of COPD and the CHIP mutations. However, several major issues with this manuscript need to be addressed.
Major issue:
- There are 9 genes were found mutated as CHIP, but the later part of this manuscript only focused on DNMT3A mutation. Why do the authors ignore all other genes? There are other genes related to methylation as well, for example, the TET2. The authors should address the reason why they designed the analysis like this. If the author still wants to focus on DNMT3A mutations only, the title and text should be more specific.
- In the introduction, the authors didn’t provide enough information explaining why linking CHIP to COPD.
- The authors did not provide detailed information about the CHIP mutation they detected. What are the criteria for CHIP mutations reported in this study? Deleterious only or all mutations? What’s the definition of CHIP positive (Table 1)? What are the 56 CHIP-associated genes (line 100)? All these essential information should be described clearly, and all detected CHIP mutations should be listed in the supplementary table with annotation.
- The methods related to this manuscript are not clear enough. Even though this manuscript is in Brief Report form, detailed material and methods should be included in the supplementary file.
- Figure 1F, in the western plot figure, the authors should mark more detailed information. It’s difficult for readers to understand the difference of the samples. Especially samples 1 and 2; samples 3 and 4; samples 5 and 6 are all showing (+/- and -/+).
Minor issues:
- Most of the gene names are not using italic style.
- The authors didn’t provide enough appropriate citations. For example, line 33-34, line 118-119, etc.
Author Response
Response is provided in the attached docx file

Round 2
Reviewer 2 Report
The new version is acceptable for publication in its current form.
Author Response
We would like to thank the reviewer for positive evaluation

Reviewer 3 Report
In the updated version, the authors addressed all my concerns for the previous version. Before the final acceptance, there are two minor issues to be fixed.
1.In the supplemental methods, "Common single-nucleotide polymorphisms with a minor allele frequency of at least 5% in either the 1000 Genome Project, Exome Variant Server, or ExAC databases were excluded.". MAF >5% is too high to exclude polymorphisms. 1% threshold was widely used to remove polymorphisms, (some more strict studies used 0.1%). And nowadays, gnomAD database is more informative for this purpose. Please double-check whether using the 5% MAF cutoff included some variants that should not be included.
2. There are still some gene IDs not using italic style, for example, Figure 1A
Author Response
Point by point response - Reviewer 3:
- In the supplemental methods, "Common single-nucleotide polymorphisms with a minor allele frequency of at least 5% in either the 1000 Genome Project, Exome Variant Server, or ExAC databases were excluded.". MAF >5% is too high to exclude polymorphisms. 1% threshold was widely used to remove polymorphisms, (some more strict studies used 0.1%). And nowadays, gnomAD database is more informative for this purpose. Please double-check whether using the 5% MAF cutoff included some variants that should not be included.
Thank you very much for this helpful suggestion. We checked all reported variants classified as CHIP mutations in our study for MAF>=1% in the population, as suggested by the reviewer using the gnomAD database. None of our reported variants that were classified as CHIP mutations had a MAF>=1%, therefore we can exclude an overestimation of somatic mutations called in our cohort.
- There are still some gene IDs not using italic style, for example, Figure 1A
As suggested by the reviewer we have modified all the gene IDs in Fig 1A, Fig 2A, Table S1, Table S2, Table S4, Table S5, also in the revised version of the manuscript.
